# Association between birth location and short-term outcomes for babies with gastroschisis, congenital diaphragmatic hernia and oesophageal fistula: a systematic review

Behrouz Nezafat Maldonado [ID],[1] Graciaa Singhal [ID],[2] LiYan Chow,[3] Dougal Hargreaves,[4] Chris Gale [ID],[1] Cheryl Battersby [ID] [1]

[1]Neonatal Medicine, Chelsea and Westminster Hospital Campus, Imperial College London, London, UK
[2]Department of Medicine, Hull York Medical School, Hull, UK
[3]Neonatal Medicine Department, Chelsea and Westminster Hospital NHS Foundation Trust, London, UK
[4]School of Public Health, Imperial College London, London, UK

**Correspondence to**
Dr Cheryl Battersby; c. battersby@imperial.ac.uk

## ABSTRACT

**Background** Neonatal care is commonly regionalised, meaning specialist services are only available at certain units. Consequently, infants with surgical conditions needing specialist care who are born in non-surgical centres require postnatal transfer. Best practice models advocate for colocated maternity and surgical services as the place of birth for infants with antenatally diagnosed congenital conditions to avoid postnatal transfers. We conducted a systematic review to explore the association between location of birth and short-term outcomes of babies with gastroschisis, congenital diaphragmatic hernia (CDH) and oesophageal atresia with or without tracheo-oesophageal fistula (TOF/OA).

**Methods** We searched MEDLINE, CINAHL, Web of Science and SCOPUS databases for studies from high income countries comparing outcomes for infants with gastroschisis, CDH or TOF/OA based on their place of delivery. Outcomes of interest included mortality, length of stay, age at first feed, comorbidities and duration of parenteral nutrition. We assessed study quality using the Newcastle-Ottawa Scale. We present a narrative synthesis of our findings.

**Results** Nineteen cohort studies compared outcomes of babies with one of gastroschisis, CDH or TOF/OA. Heterogeneity across the studies precluded meta-analysis. Eight studies carried out case-mix adjustments. Overall, we found conflicting evidence. There is limited evidence to suggest that birth in a maternity unit with a colocated surgical centre was associated with a reduction in mortality for CDH and decreased length of stay for gastroschisis.

**Conclusions** There is little evidence to suggest that delivery in colocated maternity-surgical services may be associated with shortened length of stay and reduced mortality. Our findings are limited by significant heterogeneity, potential for bias and paucity of strong evidence. This supports the need for further research to investigate the impact of birth location on outcomes for babies with congenital surgical conditions and inform future design of neonatal care systems.

**PROSPERO registration number** CRD42022329090.

### WHAT IS ALREADY KNOWN ON THIS TOPIC

⇒ Infants with congenital surgical anomalies require specialist surgical care and neonatal care. This specialist care may not be available across all units. As a result, some infants are delivered in a non-surgical centre and are then postnatally transferred to a surgical centre. International consensus recommends delivery in colocated maternity and surgical units for babies with antenatally diagnosed surgical conditions.

### WHAT THIS STUDY ADDS

⇒ We systematically identified international data exploring the impact of birth in a non-surgical centre on neonatal outcomes for infants with congenital diaphragmatic hernia, gastroschisis or TOF/OA. There is limited evidence to suggest that birth in a non-surgical centre may be associated with increased mortality, prolonged length of stay and delayed enteral feeding.

### HOW THIS STUDY MIGHT AFFECT RESEARCH, PRACTICE OR POLICY

⇒ This study highlights key gaps in the evidence; studies able to account for case-mix and clinical severity and reporting on core outcomes across different birth locations are needed to inform future development of neonatal care systems.

## INTRODUCTION

Infants with congenital surgical conditions require specialist surgical input soon after birth. It is important to ensure that specialist services are available to adequately care for these babies. Neonatal care may be delivered through centralised or decentralised models. This means that different levels of care may be available at birth across healthcare settings.

Antenatal care aims to identify congenital anomalies, for optimal delivery planning with access to specialist care. Guidelines

recommend delivery in a centre with neonatal intensive care and neonatal surgical services located within the same facility for infants with suspected surgical anomalies.[1–3] However, due to socioeconomic, environmental or health factors, or the nature of the defect anomalies are not identified antenatally, and infants are born in a unit without surgical services. Postnatal transfer can hinder parental access and bonding with their baby, as complex infants may be cared for in specialist centres far from the family residence. Clinical pathways should enable provision of intensive care to all infants where this is required, while minimising postnatal transfer.[3]

To improve neonatal surgical care, we must understand how birthplace impacts outcomes for congenital anomalies. Some anomalies, such as gastroschisis, are often diagnosed antenatally, while others such as tracheo-oesophageal fistula or oesophageal atresia (TOF/OA) are not. Other defects such as congenital diaphragmatic hernia (CDH) may be antenatally diagnosed and require intensive care from birth, meaning delivery in the right place can be key. As a result, planning the place of delivery varies based on the condition. Gastroschisis, TOF/OA and CDH have different rates of antenatal detection, and the potential benefits of being born in the right place will vary depending on the condition and the nature of the defect itself. However, they serve as exemplar congenital anomalies to explore the impact of birth location on short-term outcomes.

This review aims to understand the impact of delivery location on short-term neonatal outcomes for infants born with CDH, gastroschisis and TOF/OA. We aim to map the latest evidence to inform future research and influence neonatal service design.

## METHOD

### Search methods for identification of studies

This systematic review was prospectively registered in PROSPERO (Registration number: CRD42022329090) and conducted in accordance with Preferred Reporting Items for Systematic Reviews and Meta-Analyses (PRISMA).[4 5] We searched the following databases: MEDLINE, CINAHL, Web of Science and SCOPUS. Online supplemental file 1 includes example search strategy.

### Selection criteria

Studies were eligible for inclusion if they:
1. Were published between 1 January 2000 and 15 January 2023 in any language.
2. Were conducted in a high-income country as defined by the World Bank.[6]
3. Included a study population of infants diagnosed with one of CDH, gastroschisis and TOF/OA.
4. Provided data on birth location across at least two groups.
5. Reported at least one outcome of interest.

Studies were excluded if they did not report delivery location, did not provide quantitative data or did not allow for comparisons. We also searched the bibliography of included studies, grey literature and contacted experts for relevant papers.

### Outcomes of interest

Outcomes of interest were mortality, length of stay, time to full enteral feeds, time to first feed, sepsis, presence of morbidity, duration of parenteral nutrition, presence of liver disease, growth and breastfeeding rates (as a proxy measure for maternal-separation). Outcomes were decided on reviewing the literature, including core outcome set on gastroschisis and core outcome set in neonatology.[7 8]

### Selection of studies

Three reviewers (BNM, LC and GS) screened title and abstracts using CADIMA software for records management, deduplicate, screening and data extraction.[9] Conflicts were resolved by team discussions. Data extraction was carried out using a predetermined template to include baseline characteristics and outcome data. Two reviewers (BNM and GS) conducted data extraction independently with conflicts resolved through discussions. Only data related to CDH, gastroschisis or TOF/OA were extracted from studies involving infants with other surgical conditions.

### Risk of bias and quality of evidence assessment

The Newcastle-Ottawa Scale (NOS) assessed quality and risk of bias of included studies,[10] using nine points across three domains: (1) participants selection; (2) comparability and (3) detection of exposure/outcome. Depending on the scoring, studies were rated as 'good', 'fair' or 'poor' quality using standard guidance. The NOS assessment was independently done by two reviewers (BNM and GS) and discrepancies resolved by discussion. Studies that included more than one condition of interest but used the same methodology were assessed once. Studies that include the same population or potentially overlapping populations were independently assessed.

### Data extraction and analysis

Data extraction followed a predefined collection form. We present findings grouped by condition. For each outcome we present, when available, the absolute rates as well as unadjusted and adjusted results in cases and control, and results of statistical significance tests. We use vote counting based on the direction of effect methodology, comparing the number of positive studies—that showed benefit—to the number of negative studies—showing harm. We report direction of effect by outcome and study quality, and we also present the overall effect of birth location.[11] The author's conclusion was also considered in the overall effect.

### Patient and public involvement

No patients or members of the public were involved in this systematic review.

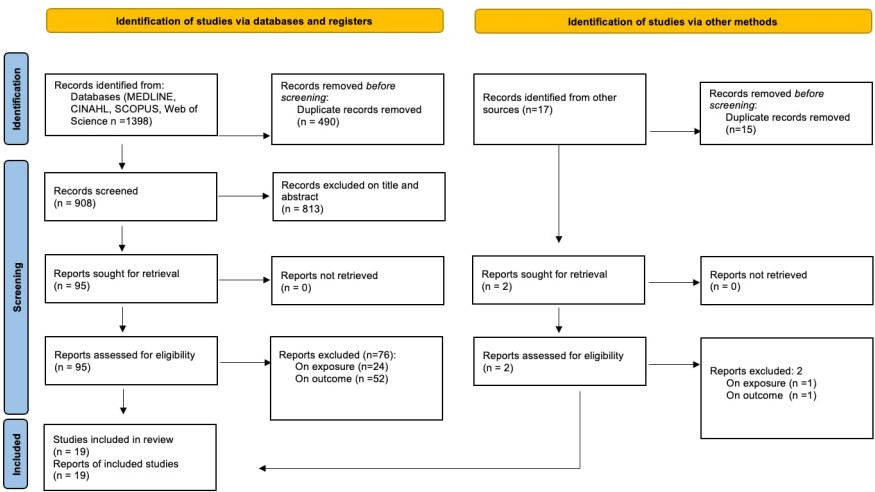

**Figure 1** Preferred Reporting Items for Systematic Reviews and Meta-Analyses flowchart.

## RESULTS

### Description of studies

Of 1398 records retrieved, 909 remained after deduplication and were screened on title and abstract. A total of 813 were excluded on title and abstract and 95 full texts were retrieved and assessed for eligibility. Nineteen studies met our inclusion criteria and were included in the review. Reasons for exclusion included the study setting, the lack of a control and exposure group or no comparison between groups provided in the results. Grey literature and bibliography searches did not yield any further references. Figure 1 shows the PRISMA flow diagram for the records found and screened.

### Study characteristics

Studies were either conducted in a single country, for example, Australia,[12–14] Canada,[15–17] France,[18] Germany,[19] Japan,[20] USA,[21–28] or at an international level across multiple countries.[29 30] One study included data on all three congenital malformations of interest for the review,[12] while the rest focused on only one of them. Study size ranged from 50 to 4663 babies, and we expect overlap in the populations included across the studies in view of the time period and settings included. Case-mix adjustments were carried out in eight studies.[12 13 15 16 22 27 29 31] The main characteristics of included studies are summarised in table 1.

Mortality was most reported; however, the definition of mortality was inconsistent across studies. Other outcomes reported included length of hospital stay, postnatal age at first feed, postnatal age at full enteral feeding, presence of comorbidities and duration of total parenteral nutrition. Only mortality was reported for all conditions, and some outcomes were not reported for certain conditions.

### Quality assessment

We assessed study quality using the NOS.[10] A composite score depending on the score across the three domains was assigned to each study, as described previously. Study quality assessment is summarised in table 2. All studies performed well in the selection and outcome domains of the tool; however, performance in the comparability domain was variable. Ten studies were rated as 'good' quality and nine as 'poor' quality.

### Protocol modifications

High degree of heterogeneity precluded a meta-analysis. Instead, we present a narrative synthesis with vote counting on the direction of effect.[11]

### Vote counting for measure of effect

We present the direction of effect for each study for the outcomes reported. In addition, we report the overall direction of effect of birth in surgical centre for CDH, gastroschisis and TOF/OA. Figure 2 displays vote counting on overall direction of effect. Table 3 summarises direction of effect per outcome and the quality rating per study.

### Congenital diaphragmatic hernia

Eleven observational studies included babies with CDH, two were prospective while nine were retrospective studies.[12 15 17 18 20–24 29 30] Three used single-centre data to compare two populations—infants born within their unit compared with infants referred from other hospital and transferred in. The other eight studies explored data across multiple centres through patient databases and used the birth location and subsequent transfers for surgical management to define two groups. Mortality was reported in 10 studies; however, the definition of mortality varied—from death during admission to death in infancy. Two studies included data from multiple countries, members of the Congenital Diaphragmatic Hernia Study Group (CDHSG).[29 30] There is potential overlap in populations across studies using patient databases; however, we were unable to distinguish the cohorts. The two studies using data from the CDHSG include the same population; however, they report different outcomes (mortality vs morbidity) and as such both studies have been included.

**Table 1** A: Summary table of included studies on CDH (see online supplemental material 1 for complete table), B: Summary table of included studies on gastroschisis (see online supplemental material 1 for complete table), C: Summary table of included studies on TOF/OA (see online supplemental material 1 for complete table)

| Author/year/setting | Population | Study Sample | Comparison | Outcomes | Results—absolute numbers | Results—analysis presented§ | Author's conclusion on the effect of birth location on outcomes ?* |
|---|---|---|---|---|---|---|---|
| Al-Shanafey et al, 2002, Halifax, Canada[17] | Infants born with CDH born in or referred to regional unit between 1973 and 1999 | 81 infants total: 41 transferred to surgical centre (outborn) | Born in specialist surgical centres versus born outside and referred | Mortality | No transfer: 9/40 died Transfer: 7/41 died | Non-significant difference between groups, p=0.54 | No difference |
| Algert et al, 2008, New South Wales, Australia[12] | Infants born alive with CDH managed in a single regional centre between 2001 and 2003 | 57 infants total: 22 transferred to surgical centre (outborn) | 'Colocated hospital'—not requiring neonatal emergency transport versus with 'other hospital'—requiring transport | 1-year mortality | No transfer: 19/35 died Transfer: 4/22 died | Significant difference between groups, p<0.01 | Favour birth in surgical centre/regionalisation of services |
| Aly et al, 2010, USA[21] | All infants with CDH and respiratory distress requiring mechanical ventilation admitted to selected hospitals during 1997–2004 | 2140 infants total: 1020 transferred to surgical centre (outborn) | Group 1 did not undergo transport versus group 2 transported from the birth hospital to the surgical facility | Mortality after surgery | No transfer: 143/1120 died Transfer: 165/1020 died | aOR: 1.46 (1.02–2.05) p=0.02 | Favour birth in surgical centre/regionalisation of services |
| Boloker et al, 2002, New York, USA[22] | Infants born with CDH and admitted to a single surgical centre during 1992–2000 | 120 infants total: 53 transferred to surgical centre (outborn) | Inborn versus outborn transfers | Mortality | No transfer: 23/67 died Transfer: 6/53 died | | No difference |
| Carmichael et al, 2020, California, USA[23] | Infants born with CDH during 2006–2011 in California | 577 infants total: 357 transferred to surgical centre (outborn) | Transfer from birth hospital for repair versus no transfer from birth hospital | Mortality to infancy | No transfer: 103/220 died Transfer: 77/357 died | Unadjusted: HR: 0.4 (0.3–0.5) Adjusted: aHR: 0.3 (0.2–0.5) | Favour birth in surgical centre/regionalisation of services |

Continued

**Table 1** Continued

| Author/year/setting | Population | Study Sample | Comparison | Outcomes | Results—absolute numbers | Results—analysis presented§ | Author's conclusion on the effect of birth location on outcomes?* |
|---|---|---|---|---|---|---|---|
| Gallot et al, 2007, Central–Eastern France[18] | Infants born with CDH during 1986–2003 | 387 infants total: 211 transferred (outborn) | Delivery in tertiary hospital versus peripheral hospital and transfer | Mortality | No transfer: 78/176 died; Transfer: 74/211 died | Group difference: p=0.06 | Favour birth in surgical centre/regionalisation of services |
| Nagata et al, 2013, Japan[20] | Infants born with CDH during 2006–2010 | 614 infants total: 165 transferred to surgical centre (outborn) | Inborn versus outborn | Mortality | No transfer: 129/449 died; Transfer: 22/165 died | Group difference: p<0.001 | N/A |
| Nasr and Langer, 2011, Canada[15] | Infants born with CDH during 2005–2008 | 140 infants total: 65 transferred (outborn) | Inborn: born in the same building as NICU versus outborn requiring an ambulance transfer | Mortality | No transfer: 17/75 died; Transfer: 21/65 | Group difference: p=0.30; aOR 2.8, p=0.04 | Favour birth in surgical centre/regionalisation of services |
| Putnam et al, 2016, Congenital Diaphragmatic Hernia Study Group (CDHSG): Australia, Canada, Chile, Germany, Italy, Japan, Malaysia, Poland, Russia, Scotland, Sweden, The Netherlands, USA[29] | Infants born with CDH during 2007–2014 | 3665 infants: number of transferred not available | Inborn versus outborn | Pulmonary morbidity; Neurologic morbidity; Gastrointestinal morbidity | 327/660 transferred and had pulmonary morbidity; 228/446 transferred and had neurologic morbidity; 787/1348 transferred and had gastrointestinal morbidity | OR 1.25 (0.86–1.84), p=0.247; OR 1.04 (0.37–1.47), p=0.848; OR 1.64 (1.11–2.42), p=0.013 | No difference |
| Sola et al, 2010, USA[24] | Infants with CDH during 1997–2006 | 2774 infants total: 891 transferred (outborn) | Hospital transfer versus birth | Mortality | No transfer: 333/1680 died; Transfer: 212/891 died | Group difference: p=0.003 | No difference |

Continued

**Table 1** Continued

| Author/year/setting | Population | Study Sample | Comparison | Outcomes | Results—absolute numbers | Results—analysis presented§ | Author's conclusion on the effect of birth location on outcomes?* |
|---|---|---|---|---|---|---|---|
| Stopenski et al, 2022, CDHSG: Australia, Canada, Chile, Germany, Italy, Japan, Malaysia, Poland, Russia, Scotland, Sweden, The Netherlands, USA[30] | Infants with CDH during 2007–2019 with prenatal diagnosis | 4195 infants total: 1108 transferred (outborn) | Inborn—delivered and treated at CDHSG hospitals. Outborn—delivered in a separate institution and required transfer to a definitive care centre. | Mortality | No transfer: 2082/3087 died. Transfer: 733/1108 died | Group difference: p=0.44 OR: 1.07 (0.015–1.25) p=0.391 | No difference |
| Algert et al, 2008, New South Wales, Australia[12] | Infants born with gastroschisis during 2001–2003 | 62 infants: 11 transferred to a surgical centre | Colocated hospital—not requiring neonatal emergency transport compared with other hospital—requiring transport | 1-year mortality | Transfer: 0/11 died. No transfer: 7/51 died | Non-significant difference | No difference |
| Dalton et al, 2017, USA[25] | Infants born with gastroschisis during 2010–2015 | 79 infants total: 18 transferred to surgical centre (outborn) | Inborn versus outborn | Length of stay‡ Days to first feed Days to full feed Total parenteral nutrition days | Transfer: 40±26 days No transfer: 30±15 days Transfer: 15±8.4 days No transfer: 13±8.5 days Transfer: 36±23 days No transfer: 16±54 days Transfer: 32±23 days No transfer: 23±12 days | Group difference: p=0.03 Group difference: p=0.17 Group difference: p=0.07 Group difference: p=0.03 | Favour birth in surgical centre/regionalisation of services |

Continued

**Table 1** Continued

| Author/year/ setting | Population | Study Sample | Comparison | Outcomes | Results—absolute numbers | Results—analysis presented§ | Author's conclusion on the effect of birth location on outcomes ?* |
|---|---|---|---|---|---|---|---|
| Hong et al, 2019, USA[26] | Infants>1500 g born with gastroschisis during 2009–2015 | 4663 infants total: 1134 transferred (outborn) | Inborn—birth in surgical centre versus outborn— delivery outside of centre | Mortality Length of stay‡ | Transfer: 23/1134 died No transfer: 67/3529 died Transfer: 38 (27,64) days No transfer: 35 (26,55) days | aOR: 1.02 (0.60–1.72) p>0.05 Poisson regression: 0.10 (0.04,0.16) p<0.05 | No difference |
| Kandasamy et al, 2009, North Queensland, australia[13] | Infants born with gastroschisis during 1988–2007 | 50 infants total: 10 transferred (outborn) | Inborn versus outborn | Mortality Length of stay | No numerical data available—no difference between groups found | | No difference |
| Kitchanan et al, 2000, North Queensland, australia[14] | Infants born with gastroschisis during 1988–1997 | 21 infants (inborn): 4 transferred (outborn) | Inborn— delivered in regional tertiary unit versus outborn— delivered in peripheral hospitals | Days to first feed Days to full enteral feeding Total parenteral nutrition days | Transfer: 25 days No transfer: 9 days Transfer: 19 days No transfer: 16 days Transfer: 42 days No transfer: 14 days | Group difference: p=0.012 Group difference: p=0.02 Group difference: p=0.012 | No difference |

Continued

**Table 1** Continued

| Author/year/setting | Population | Study Sample | Comparison | Outcomes | Results—absolute numbers | Results—analysis presented§ | Author's conclusion on the effect of birth location on outcomes ?* |
|---|---|---|---|---|---|---|---|
| Nasr and Langer, 2012, Canada[16] | Infants born with gastroschisis during 2005–2008 | 395 infants: 158 transferred | Inborn—born in the same building as NICU versus outborn—requiring an ambulance transfer | Mortality Length of stay‡ Days to first feed Total parental nutrition days Complications | Transfer: 4/158 died No transfer: 11/237 died Transfer: 49 (27,62) days No transfer: 48 (21,54) days Transfer: 12 (6,14) days No transfer: 12 (7,15) days Transfer: 42 (22,40) days No transfer: 36 (17,40) days Transfer: 4/158 No transfer: 11/237 | Group difference: p=0.10 Group difference: p=0.80 Group difference: p=0.85 Group difference: p=0.19 Group difference: p=0.10 OR 1.60 (1.09–2.7), p=0.05 | No difference |
| Savoie et al, 2014, USA[27] | Infants with gastroschisis during 2008–2013 | 524 infants total: 239 transferred outborn | Inborn—delivery in hospital where surgical repair occurred versus outborn—required transfer to surgical centre for repair | 30-day Mortality Length of stay‡ Days to first enteral feed Days to full enteral feeding Total parenteral nutrition days | Transfer: 4/239 died No transfer: 4/283 died Transfer: 42 (28,76) days No transfer: 34 (24,63) days Transfer:18 (13,26) days No transfer:16 (11,22) days Transfer: 31 (23,57) days No transfer: 27 (20,43) days Transfer: 29 (20,54) days No transfer: 27 (17,445) days | Group difference: p=0.81 Group difference: p=0.002 Group difference: p=0.004 Group difference: p=0.0008 Group difference: p=0.06 | Favour birth in surgical centre/ regionalisation of services |

**Table 1** Continued

| Author/year/ setting | Population | Study Sample | Comparison | Outcomes | Results—absolute numbers | Results—analysis presented§ | Author's conclusion on the effect of birth location on outcomes ?* |
|---|---|---|---|---|---|---|---|
| Algert et al, 2008, New South Wales, Australia[12] | Infants with oesophageal atresia during 2001–2003 | 42 infants: 25 transferred | Colocated hospital—not requiring neonatal emergency transport compared with 'other hospital'— requiring transport | 1-year mortality | Transfer: 1/25 died No transfer: 3/17 died | Non-significant difference between groups | No differences |
| Schlee et al, 2022, Germany[19] | Infants with oesophageal atresia during 2009–2013 | 57 infants total: 35 transferred (outborn) | Inborn—born in institution versus outborn— referred postnatally and transported for surgery | 1-year mortality Length of stay‡ Days to full enteral feeding | Transfer: 0/35 died No transfer: 2/17 died Transfer: 17 (13,36) days No transfer: 92 (18,158) days Transfer: 8 (6,11) days No transfer: 12 (7,18) days | Group difference: p=0.103 Group difference: p=0.009 Group difference: p=0.137 | No difference |
| Wang et al, 2014, USA[28] | Infants with oesophageal atresia/ tracheo-oesophageal fistula during 1997–2009 | 4168 infants total: 1169 transferred (outborn) | Divided by admission source: birth versus hospital transfer | Mortality | Transfer: 99/1169 died No transfer: 186/1850 died | Group difference: p=0.399 | No difference |

*Not all authors provided conclusions on favoured birth location.
†Potentially overlapping cohort: multicentre or population-based studies may include duplicates infants.
‡Length of stay given as median (IQR).
§Not all studies provided statistical analysis, that is, ORs, some studies only described the number of events.
aHR, adjusted HR; aOR, adjusted OR; CDH, congenital diaphragmatic hernia; NICU, neonatal intensive care unit; OA, oesophageal atresia; TOF, tracheo-oesophageal fistula.

**Table 2** Quality Assessment: Newcastle-Ottawa Quality Assessment Scale for cohort studies

| Author, year† | Selection | | | | Comparability | | Outcome | | | Overall rating* |
|---|---|---|---|---|---|---|---|---|---|---|
| | 1 | 2 | 3 | 4 | 5 | 6 | 7 | 8 | 9 | |
| Al-Shanafey (2002)[17] | * | * | * | * | | | * | * | * | Poor |
| Algert (2008)[12] | * | * | * | * | | | * | * | * | Poor |
| Aly (2010)[21] | * | * | * | * | * | * | * | * | * | Good |
| Boloker (2002)[22] | * | * | * | * | | | * | * | * | Poor |
| Carmichael (2020)[23] | * | * | * | * | * | * | * | * | * | Good |
| Dalton (2017)[25] | * | * | * | * | | | * | * | * | Poor |
| Gallot (2007)[18] | * | * | * | * | * | | * | * | * | Good |
| Hong (2019)[26] | * | * | * | * | * | | * | * | * | Good |
| Kandasamy (2009)[13] | * | * | * | * | | | * | * | * | Poor |
| Kitchanan (2000)[14] | * | * | * | * | | | * | * | * | Poor |
| Nagata (2013)[20] | * | * | * | * | | | * | * | * | Poor |
| Nasr (2011)[15] | * | * | * | * | * | * | * | * | * | Good |
| Nasr (2012)[16] | * | * | * | * | * | * | * | * | * | Good |
| Putnam (2016)[29] | * | * | * | * | * | | * | * | * | Good |
| Savoie (2014)[27] | * | * | * | * | * | | * | * | * | Good |
| Schlee (2022)[19] | * | * | * | * | | | * | * | * | Poor |
| Sola (2010)[24] | * | * | * | * | * | | * | * | * | Good |
| Stopenski (2022)[30] | * | * | * | * | * | * | * | * | * | Good |
| Wang (2014)[28] | * | * | * | * | | | * | * | * | Poor |

*Good quality rating for those with 3 or 4 stars in selection domain AND 1 or 2 stars in comparability domain AND 2 or 3 stars in outcome/exposure domain. Fair quality for 2 stars in selection domain AND 1 or 2 stars in comparability domain AND 2 or 3 stars in outcome/exposure domain. Poor quality for 0 or 1 star in selection domain OR 0 stars in comparability domain OR 0 or 1 stars in outcome/exposure domain.
†Cohort studies: (1) representativeness of the exposed cohort, (2) selection of the non-exposed cohort, (3) ascertainment of exposure, (4) demonstration that the outcome of interest was not present at start of the study, (5) comparability of cohorts based on the design or analysis most important factor, (6) comparability of cohorts based on the design or analysis second important factor, (7) assessment of outcome, (8) was follow-up long enough for outcomes to occur and (9) adequacy of follow-up of cohort.

Five studies with exploring mortality favoured delivery in a surgical centre for infants with CDH—three of these rated as 'good' quality studies (figure 2). The other five studies reported no difference in mortality. No difference in morbidity rate was reported in an international study from the CDHSG.[29]

We are unable to report other outcomes such as length of stay, postnatal age at first feed and duration of parenteral nutrition due to reporting limitation as these were unreported in the included studies.

Five population-level studies carried out case-mix adjustments. Three of these studies concluded that birth in surgical centre was favourable for this population, while two studies using international data did not find a difference in mortality after case-mix adjustments. We also found differences in the population included in the studies, some studies excluded infants with other congenital anomalies while other studies had no exclusion criteria—this brings selection bias and would influence the mortality rates reported. In addition, the rate of antenatal diagnosis and termination of pregnancy for congenital anomalies varies across study setting and period, which would impact the birth location and the survival rate.[32 33] It is important to highlight that some authors recognised that the reported results may be skewed by 'hidden mortality', as infants may have died at delivery or prior to surgery or reaching the surgical centre and that would have impacted on the population included in their study.

### Gastroschisis

Seven studies explored the impact of delivery site on the outcomes for neonates diagnosed with gastroschisis.[12–14 16 25–27] All studies were single country. Three studies were rated 'good' quality and four 'poor' quality. Three multicentre studies used population level databases.[16 26 27] Inclusion and exclusion criteria varied across the studies, notably one study excluded complicated gastroschisis,[25] while others did not describe whether the population study included simple or complex gastroschisis, limiting comparison of the findings.

There may be overlap across USA-based cohorts, but we were unable to distinguish them. Five studies reported on mortality and all found no differences between

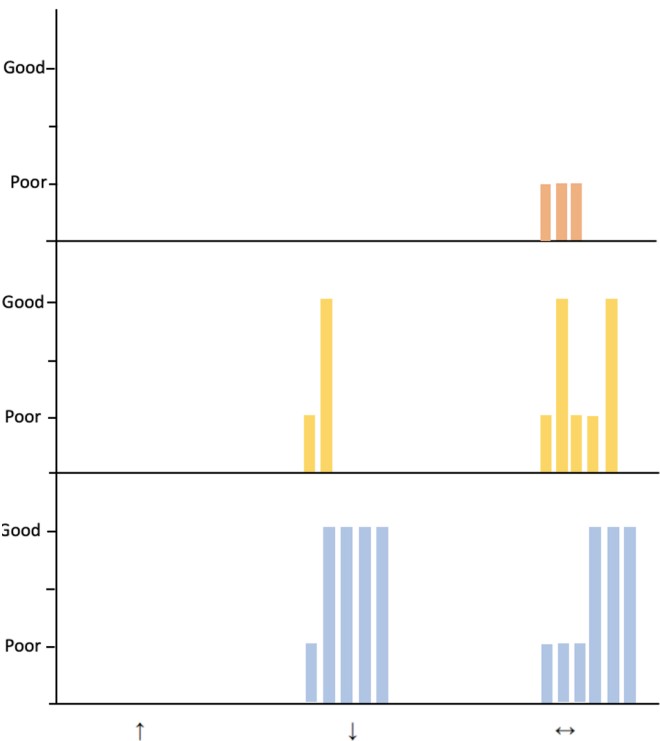

**Figure 2** Harvest plot of overall direction of effect in included studies by overall quality score. Height depicts overall quality judgement and colour depicts surgical conditions (blue=congenital diaphragmatic hernia; yellow=gastroschisis; orange=tracheo-oesophageal fistula or oesophageal atresia). Arrows denote adverse effects (↑), beneficial effects (↓) or conflicting evidence (↔) of birth in a surgical unit.

groups.[12 13 16 26 27] Median length of stay was explored in five studies. Two of these found transfers to a surgical centre were associated longer stay, these were rated as 'poor' and 'good' quality studies.[25 27] The other three studies found no difference in length of stay.[13 16 26] Four studies reported on duration of total parenteral nutrition, and one study, rated as 'poor' quality, found that those infants born outside a surgical centre had a longer duration while the others found no significant difference.[14 16 25 27] One study adjusted for clustering within hospital and found no difference in mortality or length of stay.[26] Two studies adjusted for case-mix of unspecified characteristics and one concluded that birth in a surgical centre led to favourable outcomes, particularly length of stay.[29]

### Oesophageal atresia and tracheo-oesophageal atresia

Three included studies explored the impact of birth location and outcomes for babies diagnosed with OA or tracheo-oesophageal atresia.[12 19 28] All studies used retrospective data. Algert *et al*, and Schlee *et al*, included data from a single surgical centre in Australia and Germany respectively, while Wang *et al*, used population-level data to study the population across the USA, findings showed no difference on mortality by location of birth but lower mortality in specialised children's hospitals

after excluding for immediate transfers. All studies were rated 'poor' quality. No study adjusted for case-mix in their analysis. No differences in outcomes of interest—mortality, length of stay or days to full enteral feeding—were found across the studies included.

### DISCUSSION

This systematic review identified 19 studies exploring the impact of birth location on short-term outcomes of infants born with gastroschisis, CDH or OA and tracheo-oesophageal atresia. All studies were observational cohort studies using data from high income countries to explore the impact of birth location on short-term outcomes. Although we aimed to report on different short-term outcomes, mortality was the only outcome that we could report for all three condition due to the data available. Study quality and risk of bias was assessed using the Newcastle-Ottawa Scale. Case-mix adjustment, not included within the quality rating, was not carried out in all studies. High study heterogeneity precluded a meta-analysis. We present a narrative synthesis with vote counting on the direction of effect. The overall direction of effect suggests limited evidence that birth in a non-surgical centre may be associated with worse outcomes particularly for increased mortality for CDH and prolonged length of stay in gastroschisis, with conflicting evidence for other outcomes and TOF/OA (figure 2). No studies explored the impact of birth location on maternal-baby separation, and impact on breastfeeding, skin to skin or family integrated care, despite the positive impact of family integrated care on neonatal outcomes.[34]

We found high heterogeneity across all included studies, and findings need to be interpreted with caution. Across all three conditions, fetal and neonatal variables, for example, associated congenital anomalies, comorbidities, surgical techniques and surgical complications may impact on outcomes and were either not reported or not adjusted for. In addition, the time of diagnosis, the nature of the defect whether simple or complex and the size of the defect may impact both the location of birth as well as the short-term outcomes. The included population also varied, for example, studies exploring CDH, may have included complex babies with severe cardiac anomalies or genetic conditions. Complex babies are more likely to be delivered in centres with colocated services, increasing the risk of bias in our findings.[12] In addition, outcome reporting was inconsistent, limiting our ability to draw conclusions—for example, some studies reported mortality to discharge home while other reported mortality to 1 year of age.[35] We included studies across a 20-year period, during which there have been significant improvements in fetal medicine, neonatology and surgery. This is reflected in the changes in survival rates during this time, for example, CDH mortality rates have decreased by 2.4% annually since 2001.[36–39] This introduces a time period bias when comparing the results.

**Table 3** Summary of direction of effect of birth in a surgical unit

| Author/year | Mortality | Length of stay | Age first enteral feed | Days to full enteral feeding | PN duration | Morbidity | Overall direction of effect (D) of birth in surgical centre |
|---|---|---|---|---|---|---|---|
| **Congenital diaphragmatic hernia** | | | | | | | |
| Al-Shanafey et al, 2002[17] | ◄► | | | | | | ◄► |
| Algert et al, 2008[12] | ▼ | | | | | | ▼ |
| Aly et al, 2010[21] | ▼* | | | | | | ▼* |
| Boloker et al, 2002[22] | ◄► | | | | | | ◄► |
| Carmichael et al, 2020[23] | ▼* | | | | | | ▼* |
| Gallot et al, 2007[18] | ▼ | | | | | | ▼ |
| Nagata et al, 2013[20] | ◄► | | | | | | ◄► |
| Nasr and Langer, 2011[15] | ▼* | | | | | | ▼* |
| Putnam et al, 2016[29] | | | | | | ◄►* | ◄►* |
| Sola et al, 2010[24] | ◄► | | | | | | ◄► |
| Stopenski et al, 2022[30] | ◄►* | | | | | | ◄►* |
| **Gastroschisis** | | | | | | | |
| Algert et al, 2008[12] | ◄► | | | | | | ◄► |
| Dalton et al, 2017[25] | | ▼ | ▼ | ▼ | ▼ | | ▼ |
| Hong et al, 2019[26] | ◄►* | ◄►* | | | | | ◄►* |
| Kandasamy et al, 2009[13] | ◄► | ◄► | | | | | ◄► |
| Kitchanan et al, 2000[14] | | | ▼ | ▼ | ▼ | | ◄►* |
| Nasr and Langer, 2012[16] | ◄►* | ◄►* | ◄►* | ◄►* | ◄►* | | ◄►* |
| Savoie et al,[27] | ◄►* | ▼* | ▼* | ▼* | ▼* | | ▼* |
| **Oesophageal atresia/tracheo-oesophageal fistula** | | | | | | | |
| Algert et al, 2008[12] | ◄► | | | | | | ◄► |
| Schlee et al, 2022[19] | ◄► | ▼ | | ◄► | | | ◄► |
| Wang et al, 2014[28] | ◄► | | | | | | ◄► |

Outcomes of interest and the overall direction of effect of birth in a surgical centre compared with birth outside a surgical centre (D) showed for each study included.
Arrows denote harm (▲), benefit (▼) or conflicting evidence (◄►).
Red colour—'poor' quality, yellow colour—'fair' quality, green colour—'good' quality.
*denotes studies carried out case-mix adjustments.

Overall, we found conflicting evidence regarding the importance of location of birth for congenital surgical anomalies. There is limited evidence to suggest that birth outside a surgical centre may be associated with worse neonatal outcomes, particularly for infants with CDH which carries a high mortality rate in the first few days after birth. Similarly, requiring transfer for surgical repair of a defect such as gastroschisis may lead to delay in establishing enteral feeding and prolonged hospital stays, as found in some of the studies included. However, it is important to recognise that delivery of infants with congenital surgical condition in centres with paediatric surgery services may also have disadvantages for families. For example, evaluation of the centralisation of congenital cardiac services in England has identified trade-offs between improved clinical outcomes and family experience—with reorganisation of care potentially widening inequalities and impacting access, with increased travel times and costs.[40]

Strengths of this systematic review include the comprehensive search criteria to yield international literature from high-income countries, with no limitations on language of publication and prospective registration. In addition to published literature, we also other sources of literature and consulted experts for additional studies. We present all outcomes reported in the included studies to reduce selective outcome reporting bias. Limitations of this review include the degree of bias discussed as well as not being able to carry out a meta-analysis of the studies included. The studies included, particularly for gastroschisis and TOF/OA, report on a population from nearly ten years ago and does reflect contemporary care

which has since advanced. For gastroschisis, for examples, bedside sutureless umbilical closure without intubation has been shown to be a feasible initial management that reduces time to gastroschisis closure.[41] Similarly, minimally invasive surgery for the repair of long gap OA may also bring about improvements in outcomes for TOF/OA. For CDH, fetoscopic endoluminal tracheal occlusion may be used to treat severe cases of CDH in utero.[42 43] This introduces a time bias as well as a selection bias to our results and the findings may not be generalisable. In addition, the lack of granular details on the severity or complexity of the condition and antenatal and postnatal interventions can make it difficult to conduct valid comparisons. Most studies report on transfer or outborn status (commonly requiring an ambulance transfer after delivery), as the birth location exposure. However, these terms are not consistently defined and may mean different things depending on setting and configuration of services. The rate of antenatal diagnosis was poorly reported throughout the studies. As a result, we are unable to separate those infants that were antenatally diagnosed and born in a surgical centre, versus those that had an antenatal diagnosis, but birth occurred outside a surgical centre (which may be planned for small defects, or unplanned), versus those that were not antenatally diagnosed. Case-mix adjustments is key to making valid and informative comparisons, due to the spectrum of disease complexity across these three conditions, for example simple versus complex gastroschisis or large defects to smaller CDH, for example.

Future research is needed to yield concrete evidence on the impact of birth outside a surgical centre on infants with congenital surgical conditions. This should include studies across different spectrums of disease severity and explore delivery in different location from high volume centres with maternity on-site, to high volume without maternity services as well low volume centres and births outside of a surgical centre. Using established routine or registry-based datasets may be the most efficient way of capturing whole-population data to explore this relationship. Analysis should adjust for case-mix, including maternal and infant characteristics. Additional outcomes, for example, maternal–baby separation, as well as long-term outcomes should also be explored. Qualitative work exploring the experience of parents and carers of infants that undergo transfers postnatally will complement quantitative work and provide strong evidence to shape future design of neonatal health services.

In conclusion, we found limited evidence suggesting that birth in a non-surgical centre may contribute to prolonged length of stay and increase mortality. However, the quality of evidence was insufficient to make concrete conclusions and further research is needed.

**Contributors** BNM, DH, CG and CB conceived and designed the study. BNM designed and carried out the search strategy. BNM, GS and LC reviewed search results, screened title/abstracts and full texts and extracted data from included studies. BNM and GS interpreted the results. BNM wrote the first draft and all authors contributed to the final manuscript. All authors have approved the final manuscript as submitted. BNM acts as guarantor for this publication.

**Funding** This research is support by the National Institute for Health Research grant ACF-2020-21-011 awarded to BNM. CG was supported by the United Kingdom Medical Research Council through a Transitional Support Fellowship (MR/V036866/1). CB is supported through a UK NIHR Advanced Fellowship personal award (NIHR300617). Imperial College London Open Access Fund supported the publication and dissemination of this work.

**Competing interests** No competing interests.

**Patient and public involvement** Patients and/or the public were not involved in the design, or conduct, or reporting, or dissemination plans of this research.

**Patient consent for publication** Not applicable.

**Ethics approval** As this was a review of available literature, no ethics approval was necessary.

**Provenance and peer review** Not commissioned; externally peer reviewed.

**Data availability statement** No data are available.

**ORCID iDs**
Behrouz Nezafat Maldonado http://orcid.org/0000-0002-7488-5564
Graciaa Singhal http://orcid.org/0000-0001-6729-5622
Chris Gale http://orcid.org/0000-0003-0707-876X
Cheryl Battersby http://orcid.org/0000-0002-2898-553X

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
