## [Reviewer comments · BMJ Paediatrics Open]

ARTICLE DETAILS

TITLE (PROVISIONAL)	Association between birth location and short-term outcomes for babies with gastroschisis, congenital diaphragmatic hernia and oesophageal fistula: a systematic review
AUTHORS	Nezafat Maldonado, Behrouz Singhal, Graciaa Chow, LiYan Hargreaves, Dougal Gale, Chris Battersby, Cheryl

VERSION 1 - REVIEW

REVIEWER	Dr. Mary Patrice Eastwood Northern Ireland Department of Health, Department of Paediatric Surgery
REVIEW RETURNED	16-Apr-2023

GENERAL COMMENTS	This systematic review was well conducted with use of PRISMA guidelines, PROSPERO registration and the search strategy well documented. Note should be made that it was difficult to report on length of stay, age at first feed, comorbidities, TPN use due to study heterogeneity. Main reporting throughout is only on mortality. Wording should be adjusted to reflect this throughout. Patients with correctable congenital malformations will require transfer to a centre with paediatric surgical capabilities as stated. The reality therefore is it unrealistic to suggest minimising postnatal transfer in many of these circumstances – particularly in the UK. The authors explain the results relating to three conditions. Each condition can be sub-divided into simple and complex depending on fetal/ neonatal factors. This is alluded to at various points and explored in depth in many of the papers cited. It should be more clearly stated. CDH patients with a small diaphragmatic defect cannot be considered in the same light as patients with a large (type C/D) defect. These may have undergone fetal intervention (FETO - TOTAL trial 2021), have significant pulmonary hypertension and have prenatally predicted worse postnatal outcomes. This should be noted in the discussion, with reference made to the CDH study group work. The CDH Study group papers (Putman and Stopenski) will have significant overlap in time-frame therefore I do not think you can include both– just the more recent one (Stopenski 2022). From the 11 CDH papers included your conclusion was 5 favour birth in surgical centre. Results should state you couldn't compare at other outcome factors such as length of stay, age at first feed, comorbidities, TPN due to reporting limitations. Stopenski 2022 comment that delivery at an ECMO centre optimises
---

	postnatal care. However, all patients entered are from tertiary referral centers. This and the defect grading should be included in discussion. In Sola et al 2010, mortality is higher in the transferred group (0.003). Gastroschisis Again, authors should take into account complex vs. simple gastroschisis being different conditions. Hong et al, 2019 – actually reports on low and high volume centres. High volume centres have more complex patients but similar mortality and length of stay. Furthermore, Algert et al 2008 excludes those antenatally diagnosed (?increased complexity), Dalton et al 2017 excludes complicated patients. Please note this more clearly. The 7 papers included the most recent cohort was only up to 2015 and this should be noted in discussion. In the discussion mention should be made of the bedside closure technique (Pet 2017) with is thought to reduce need for intubation and silo placement (for initial management of the gastroschisis) both of which require surgeons on site. OA/ TOF Papers are relatively old and care of this condition has changed in the last 10 years particularly in relation to long gap oesophageal atresia and minimally invasive surgical techniques (Patkowski 2022, van der Zee 2015). This should be noted in the discussion. The largest paper (Wang et al 2010) notes lower mortality in children dedicated hospitals when excluding hospital transfers ($p=0.008$). Text should be adjusted to reflect this. Overall, this is a well conducted study but the conclusions should be modified to reflect the varying spectrum of disease seen in different volume centres. Centralisation of care should not be confused with birth in a non-surgical centre in the discussion. Surgical conditions will require definitive surgical management and therefore transfer from a non-surgical centre if there is to be survival to discharge.
--	--

REVIEWER	Dr. Claire Granger Newcastle University
REVIEW RETURNED	21-Apr-2023

GENERAL COMMENTS	I commend the authors for this study: this is a challenging topic and the question merits ongoing assessment. However, I have some concerns about study design, and the ability to apply the conclusions drawn. Major points: These three conditions are very heterogeneous, and it is difficult to necessarily apply a blanket statement about outcomes: the potential benefit of being born in a surgical centre may be very different if your condition is a CDH, versus a TOF/OA, versus a gastroschisis. It is particularly difficult to group CDH with the other outcomes. In addition, I think it is important to explain in the methodology why these specific conditions have been chosen, whilst other complex congenital surgical anomalies have been excluded, for example other abdominal wall defects such as exomphalos. The other aspect not addressed by the authors is the bias incurred between the three groups of 1. Intentionally born in a centre without co-located surgical services. 2. Antenatally diagnosed but unexpectedly presenting to a non-surgical centre.
--

	3. Not-antenatally diagnosed. Some of the included studies specifically excluded postnatally diagnosed cases, but not all of them. There is a significantly different trajectory of postnatally diagnosed CDH, especially as these can be small defects which often have a good postnatal course, compared with an undiagnosed TOF/OA (lower rate of antenatal detection). The authors should, in their included studies, more comprehensively describe what they mean by “co-located”. Does this specifically exclude any ambulance transfer, or can co-located mean pathways where infants are born in a tertiary NICU partnered with a specified quaternary surgical centre (with no inborn infants), stabilised then transferred? These infants may be defined as “not co-located” but these pathways are well established and the initial stabilisation very practiced and anticipated, so the expectation would be that these infants would do “better” than those stabilised in a SCU/LNU. What about centres where the pathway is such that infants are born in maternity, stabilised in NICU then transferred to PICU (within the same hospital but different department)? The variation between these different pathways make it very hard to compare the outcomes of these different conditions and infants. Minor points: Page 5 line 41: should read “one study”. Table 1 is misleading and should be better clarified (specifically relating to the study sample column): with the numbers indicating infants transferred to surgical centre, does this mean specifically that the others were inborn, or not transferred? Page 14 line 18: currently reads “three studies of these studies” – should read, I think “three of these studies”.
--	--

VERSION 1 – AUTHOR RESPONSE

Reviewer 1: Dr. Mary Patrice Eastwood

Note should be made that it was difficult to report on length of stay, age at first feed, comorbidities, TPN use due to study heterogeneity. Main reporting throughout is only on mortality. Wording should be adjusted to reflect this throughout.

Response: We have added a sentence on the results (page 4 line 48) and discussion (page 15 line 8) to clarify that although we set off to report on different short-term outcomes, only mortality was reported for all three conditions.

Patients with correctable congenital malformations will require transfer to a centre with paediatric surgical capabilities as stated. The reality therefore is it unrealistic to suggest minimising postnatal transfer in many of these circumstances – particularly in the UK.

Response: We understand that in many settings minimising postnatal transfer may not be possible, depending on what services are available. However, we aim to explore the importance of co-location of maternity services and paediatric surgery services as there is a gap in evidence in this area as highlighted in the Paediatric Surgery GIRFT report.

The authors explain the results relating to three conditions. Each condition can be sub-divided into simple and complex depending on fetal/ neonatal factors. This is alluded to at various points and explored in depth in many of the papers cited. It should be more clearly stated.

Response: We have clarified this in the discussion (page 15 line 22)

CDH patients with a small diaphragmatic defect cannot be considered in the same light as patients with a large (type C/D) defect. These may have undergone fetal intervention (FETO - TOTAL trial 2021), have significant pulmonary hypertension and have prenatally predicted worse postnatal outcomes. This should be noted in the discussion, with reference made to the CDH study group work.

Response: We have included a paragraph to expand on the impact of severity of disease and the importance of reporting clear inclusion criteria or carrying out case-mix adjustment to mitigate these confounders in the discussion (page 16 lines 49-54)

The CDH Study group papers (Putman and Stopenski) will have significant overlap in time-frame therefore I do not think you can include both— just the more recent one (Stopenski 2022).

Response: Table 1 details that the populations in these two studies are overlapping. However, they report on different outcomes. Putnam reports on comorbidities for the inborn and outborn groups of the CDH Study group data whilst Stopenski reports on mortality. As they report on different outcomes we have included both studies in the review as they bring different data. We have added a sentence to detail this rationale (page 15 line 10).

From the 11 CDH papers included your conclusion was 5 favour birth in surgical centre. Results should state you couldn't compare at other outcome factors such as length of stay, age at first feed, comorbidities, TPN due to reporting limitations.

Response: We have included a sentence to detail this (page 14 line 16)

Stopenski 2022 comment that delivery at an ECMO centre optimises postnatal care. However, all patients entered are from tertiary referral centers. This and the defect grading should be included in discussion.

Response: As ECMO availability is not reported in other papers and does not fall as the exposure in our review question, we have not discussed the role of ECMO in CDH. We have included comments on the importance of defect grading both antenatally and postnatally.

In Sola et al 2010, mortality is higher in the transferred group (0.003).

Response: We note on Table 1 that mortality is higher in the transferred group in terms of absolute numbers and the p value, however as this is not adjusted for different characteristics and not deemed significant by the authors in their narrative we have given them a conflicting evidence rating (Table 3) on the vote counting. This follows best practice on vote counting methodology by Cochrane, that advises against vote counting based on statistical significance – excerpt “To undertake vote counting properly the number of studies showing harm should be compared with the number showing benefit, regardless of the statistical significance or size of their results”

Again, authors should take into account complex vs. simple gastroschisis being different conditions. Hong et al, 2019 – actually reports on low and high volume centres. High volume centres have more complex patients but similar mortality and length of stay.

Response: We have included a sentence to detail the differences between complex and simple gastroschisis (page 14 line 35).

Furthermore, Algert et al 2008 excludes those antenatally diagnosed (?increased complexity),

Response: Algert excludes infants with antenatally diagnosed lethal conditions, not with antenatally diagnosed gastroschisis/CDH/TOF/OA

Dalton et al 2017 excludes complicated patients. Please note this more clearly.

Response: This is included both in Supplementary Table 1 and in the main text (page 14, line 34)

The 7 papers included the most recent cohort was only up to 2015 and this should be noted in discussion.

Response: We have highlighted this in the discussion (page 16 line 50)

In the discussion mention should be made of the bedside closure technique (Pet 2017) with is thought to reduce need for intubation and silo placement (for initial management of the gastroschisis) both of which require surgeons on site.

Response: This has been included in the discussion (page 15, line 49)

Papers are relatively old and care of this condition has changed in the last 10 years particularly in relation to long gap oesophageal atresia and minimally invasive surgical techniques (Patkowski 2022, van der Zee 2015). This should be noted in the discussion.

Response: We have added a sentence on the advances since the included studies – thank you for providing those references (page 15 line 50)

The largest paper (Wang et al 2010) notes lower mortality in children dedicated hospitals when excluding hospital transfers ($p=0.008$). Text should be adjusted to reflect this.

Response: We have clarified this on the main text (page 14 line 54)

Centralisation of care should not be confused with birth in a non-surgical centre in the discussion. Surgical conditions will require definitive surgical management and therefore transfer from a non-surgical centre if there is to be survival to discharge.

Response: We have changed the terms used (page 15, line 39) when discussing the potentially trade-offs between minimising transfer and hence delivery in certain centres and family experience.

Reviewer 2: Dr. Claire Granger

These three conditions are very heterogeneous, and it is difficult to necessarily apply a blanket statement about outcomes: the potential benefit of being born in a surgical centre may be very different if your condition is a CDH, versus a TOF/OA, versus a gastroschisis. It is particularly difficult to group CDH with the other outcomes. In addition, I think it is important to explain in the methodology why these specific conditions have been chosen, whilst other complex congenital surgical anomalies have been excluded, for example other abdominal wall defects such as exomphalos.

Response: We have included sentences throughout the main text to clarify that variable disease severity limits the comparison that can be made.

In terms of the choice of these conditions, we have added an extra sentence to page 3 line 22 to clarify that we aim to explore the question of impact of birth location on outcomes for these three conditions serving as exemplary congenital conditions requiring surgery in the postnatal period .

The other aspect not addressed by the authors is the bias incurred between the three groups of

1. Intentionally born in a centre without co-located surgical services.
2. Antenatally diagnosed but unexpectedly presenting to a non-surgical centre.
3. Not-antenatally diagnosed.

Some of the included studies specifically excluded postnatally diagnosed cases, but not all of them. There is a significantly different trajectory of postnatally diagnosed CDH, especially as these can be small defects which often have a good postnatal course, compared with an undiagnosed TOF/OA (lower rate of antenatal detection).

Response: We have included sentences throughout the main text to clarify that variable disease severity limits the comparison that can be made (page 17 lines 7-10).

The authors should, in their included studies, more comprehensively describe what they mean by “co-located”. Does this specifically exclude any ambulance transfer, or can co-located mean pathways where infants are born in a tertiary NICU partnered with a specified quaternary surgical centre (with no inborn infants), stabilised then transferred? These infants may be defined as “not co-located” but these pathways are well established and the initial stabilisation very practiced and anticipated, so the expectation would be that these infants would do “better” than those stabilised in a SCU/LNU.

What about centres where the pathway is such that infants are born in maternity, stabilised in NICU then transferred to PICU (within the same hospital but different department)?

The variation between these different pathways make it very hard to compare the outcomes of these different conditions and infants.

Response: We have included a sentence to explain that the terminology used in the included studies regarding the birth location is not always clearly defined. To aid this, we have included a new column detailing the comparison group on the studies(Table 1a, 1b and 1c).

Regarding the cases of tertiary NICU with partnered quaternary surgical or CDH managed in PICU in some centres we agree that there is variation in pathways depending on local services and current pathways described in the studies are not clear, which makes it difficult to compare outcomes. However, we believe this is something that should be explored further in the future. There is data linkage working in England between neonatal units and PICU that will aid to understand better current pathways in place.

Page 5 line 41: should read “one study”.

Response: Thank you, this has been amended.

Table 1 is misleading and should be better clarified (specifically relating to the study sample column): with the numbers indicating infants transferred to surgical centre, does this mean specifically that the others were inborn, or not transferred?

Response: Those not transferred were inborn, we have made changes to the table to make this clear by adding a column that explains what the comparison groups are.

Page 14 line 18: currently reads “three studies of these studies” – should read, I think “three of these studies”.

Response: Thank you, this has been amended.

VERSION 2 – REVIEW

REVIEWER	Dr. Claire Granger Newcastle University
REVIEW RETURNED	03-Jun-2023

GENERAL COMMENTS	Thank you. My questions and points have been addressed comprehensively by the authors. This study is limited in its generalisability, but this has been acknowledged by the authors, and is an important question for further research.
--

REVIEWER	Dr. Mary Patrice Eastwood Northern Ireland Department of Health, Department of Paediatric Surgery
REVIEW RETURNED	05-Jun-2023

GENERAL COMMENTS	Thank-you for the amendments to the manuscript in line with the review.
---

VERSION 2 – AUTHOR RESPONSE

N/A